# Twofold symmetry of *c*-axis resistivity in topological kagome superconductor CsV$_3$Sb$_5$ with in-plane rotating magnetic field

Ying Xiang[1,4], Qing Li[1,4], Yongkai Li[2,3,4], Wei Xie[1], Huan Yang [1✉], Zhiwei Wang [2,3✉], Yugui Yao [2,3] & Hai-Hu Wen [1✉]

In transition metal compounds, due to the interplay of charge, spin, lattice and orbital degrees of freedom, many intertwined orders exist with close energies. One of the commonly observed states is the so-called nematic electron state, which breaks the in-plane rotational symmetry. This nematic state appears in cuprates, iron-based superconductor, etc. Nematicity may coexist, affect, cooperate or compete with other orders. Here we show the anisotropic in-plane electronic state and superconductivity in a recently discovered kagome metal CsV$_3$Sb$_5$ by measuring *c*-axis resistivity with the in-plane rotation of magnetic field. We observe a twofold symmetry of superconductivity in the superconducting state and a unique in-plane nematic electronic state in normal state when rotating the in-plane magnetic field. Interestingly these two orders are orthogonal to each other in terms of the field direction of the minimum resistivity. Our results shed new light in understanding non-trivial physical properties of CsV$_3$Sb$_5$.

[1] National Laboratory of Solid State Microstructures and Department of Physics, Collaborative Innovation Center of Advanced Microstructures, Nanjing University, Nanjing 210093, China. [2] Key Laboratory of Advanced Optoelectronic Quantum Architecture and Measurement, Ministry of Education, School of Physics, Beijing Institute of Technology, Beijing 100081, China. [3] Micronano Center, Beijing Key Lab of Nanophotonics and Ultrafine Optoelectronic Systems, Beijing Institute of Technology, Beijing 100081, China. [4] These authors contributed equally: Ying Xiang, Qing Li, Yongkai Li. ✉email: huanyang@nju.edu.cn; zhiweiwang@bit.edu.cn; hhwen@nju.edu.cn

Materials with a kagome lattice structure can host a rich variety of exotic states including spin liquid[1–3], spin density wave[4], charge density wave (CDW)[5,6], and superconductivity[4,5,7,8]. In addition, these kagome materials provide a novel platform to investigate topological electronic states[9–15]. Recently, a new family of kagome metals $AV_3Sb_5$ ($A$ = K, Rb, or Cs) has been discovered[16], and shortly afterwards, superconductivity is reported in this system[17–19]. The superconducting transition temperature ($T_c$) in $CsV_3Sb_5$ can be easily enhanced by the applied pressure[20–23]. Although the superconductivity is argued to be a strong-coupling one[24–26] probably with triplet pairing[27], the gap symmetry of this superconductor remains controversial[21,24–26,28–32] and requires further study. Besides the superconducting state, there is also a CDW transition in $AV_3Sb_5$[16–19]. The CDW order has been investigated experimentally[33–36] and theoretically[37–40]; it is probably driven by the nesting of saddle points near M points[18,35,39]. The electronic states of CDW are also investigated on the atomic scale[24,25,41–44]. Furthermore, the existence of $Z_2$ topologically non-trivial states in $AV_3Sb_5$ has been evidenced by the observation of symmetry-protected Dirac crossing bands[17,19,45,46], giant anomalous Hall effect[47,48], a chiral charge order[41], and the possible existence of Majorana zero mode in the vortex core[25].

The nematic electronic state breaks the symmetry of the crystal structure in many strongly correlated electron systems[49], including cuprates[50,51], iron-based superconductors[52,53], ultraclean quantum Hall systems[54], $Sr_3Ru_2O_7$[55], etc. The twofold in-plane electronic anisotropy breaks the symmetry of the underlying lattice in these materials. Superconductivity with twofold symmetry seems to be a common feature in topological superconductors. Such feature is observed by different kinds of measurements in doped $Bi_2Se_3$[56–61] and heterostructures constructed by $Bi_2Te_3$ and high-temperature superconductors[62,63]. The nematic superconductivity is explained theoretically as a consequence of superconducting order parameter with odd parity derived from the spin−orbital coupling and the multi-orbital effect in these materials[64,65].

In this work, by measuring the $c$-axis resistivity ($\rho_c$) using a Corbino-shape-like electrode configuration with in-plane rotating magnetic field, we observe a twofold rotational symmetry of angular dependent $\rho_c(\theta)$ both in the superconducting state and the normal state of the topological kagome metal $CsV_3Sb_5$. The field direction for the minimum resistance in the superconducting state is along one pair of crystalline axes of the lattice ($a$-axis), which suggests a superconducting gap maximum in this direction. By applying a very strong in-plane magnetic field, we also observe a twofold rotational symmetry of $\rho_c(\theta)$ curves in the normal state but with an orthogonal direction of the minimum resistivity comparing to that in the superconducting state. The twofold rotational symmetry of $\rho_c(\theta)$ curves may be related to the in-plane nematic electron state with the assistance of strong magnetic fields. Furthermore, we find a six-fold oscillation component in the $\rho_c(\theta)$ curve, which is due to the six-fold symmetry of the lattice. These findings contribute to a better understanding of the electronic state and the superconductivity in this topological kagome metal.

## Results

### Experimental configuration and superconducting characterization.
In order to detect the possible in-plane electronic anisotropy of the topological kagome metal $CsV_3Sb_5$, we measure the $c$-axis resistivity by using a Corbino-shape-like electrode configuration (Fig. 1a and Supplementary Fig. 3). The advantage of using this configuration is that a major part of the current is flowing along the $c$-axis, and thus is always perpendicular to the

in-plane rotating magnetic field. This would avoid the undesired angle dependence of the in-plane resistivity due to the flux flow if the current were applied along $ab$-plane. Thus, the observed angle-dependent variation of the $c$-axis resistivity with the in-plane rotating field can be safely attributed to some anisotropic electronic property in the material. The superconducting transition of the $CsV_3Sb_5$ single crystal is characterized by magnetization measurements (Fig. 1b), and the onset transition temperature is about 3.5 K determined from the enlarged view shown in the inset of Fig. 1b.

Figure 1c shows the temperature dependence of the in-plane ($\rho_{ab}$) and the $c$-axis resistivity. The normal-state resistivity shows a large anisotropy of $\alpha = \rho_c/\rho_{ab} = 23$ at 8 K, which suggests considerable two-dimensionality of the material. The CDW transition can be clearly seen at about 95 K as an anomaly of resistivity. However, there is an obvious step-like increase in the $\rho_c(T)$ curve with decreasing temperature before the drop of resistivity. This feature is different from that measured by the in-plane resistivity which only exhibits a monotonic drop crossing the CDW transition. This difference has also been observed in the sister compound of $RbV_3Sb_5$[19]. Figure 1d, e shows the temperature dependence of in-plane and $c$-axis resistivity, respectively; they are measured near the superconducting transition under different magnetic fields. The superconducting feature starts at about 3.5 K in $\rho_{ab}(T)$ at zero applied field, and the superconductivity can be easily killed by a field of about 0.8 T at 2 K. All the results of $\rho_{ab}(T)$ are similar to those in previous reports[21,29]. In the $\rho_c(T)$ curve measured at 0 T, the detected zero-resistance temperature is the same as that obtained in the $\rho_{ab}(T)$ curve. However, a superconducting-fluctuation-like behaviour can be seen obviously in the $\rho_c(T)$ curve at temperatures up to about 5.5 K. Furthermore, the resistivity drop at 2 K can be even seen under 7 T. This contrasting behaviour between $\rho_{ab}(T)$ and $\rho_c(T)$ near $T_c$ is quite interesting and deserves further study.

### Angular dependent of $c$-axis resistivity.
During the in-plane rotation of the magnetic field, the initial field direction ($\theta = 0°$) is set to be parallel to one of the sample edges (Fig. 1a), and it is found that this direction is just along one pair of in-plane crystallographic axes of the single crystal determined by Laue diffraction (see Supplementary Note 2). The angle-dependent resistivity at 2 K and different magnetic fields (Fig. 2a) shows obvious twofold symmetry. At a field below 2.4 T, $\rho_c(\theta)$ curves show local minima near $\theta = 0°$ or 180° which is in the direction of one of the principal axes, namely $a$-axis. Since the resistivity minimum touches zero in the $\rho_c(\theta)$ curve measured at 0.4 T, the twofold symmetry of $\rho_c(\theta)$ curves is supposed to be induced by the anisotropic properties of the superconducting state. Here, on one particular curve of angle-dependent resistivity, the minimum resistivity reflects a relatively larger upper critical field ($\mu_0H_{c2}$). A simple consideration based on the Ginzburg−Landau theory and the Pippard definition of the coherence length $\xi \approx \hbar v_F/\pi\Delta$ tells that $\mu_0H_{c2} \propto \Delta^2/v_F^2$ with $\hbar$ the reduced Planck constant, $\Delta$ the superconducting gap, and $v_F$ the Fermi velocity[60]. Therefore, the gap maximum may be along $a$-axis, which suggests the possible existence of a twofold symmetry of superconductivity in $CsV_3Sb_5$.

In contrast, when the magnetic field exceeds 2.4 T, the field orientation corresponding to the minimum resistivity becomes roughly orthogonal to that of superconducting state (below 2.4 T), i.e., $\rho_c(\theta)$ near the angles for the minimum of resistivity below 2.4 T now shows local maximum above 2.4 T instead (Fig. 2a). This contrasting behaviour can be easily seen in the polar illustrations in Fig. 2b, c for the curves measured at fields below and above 2.4 T, respectively. Although the $\rho_c(T)$ curve shows superconducting-fluctuation-like behaviour at fields

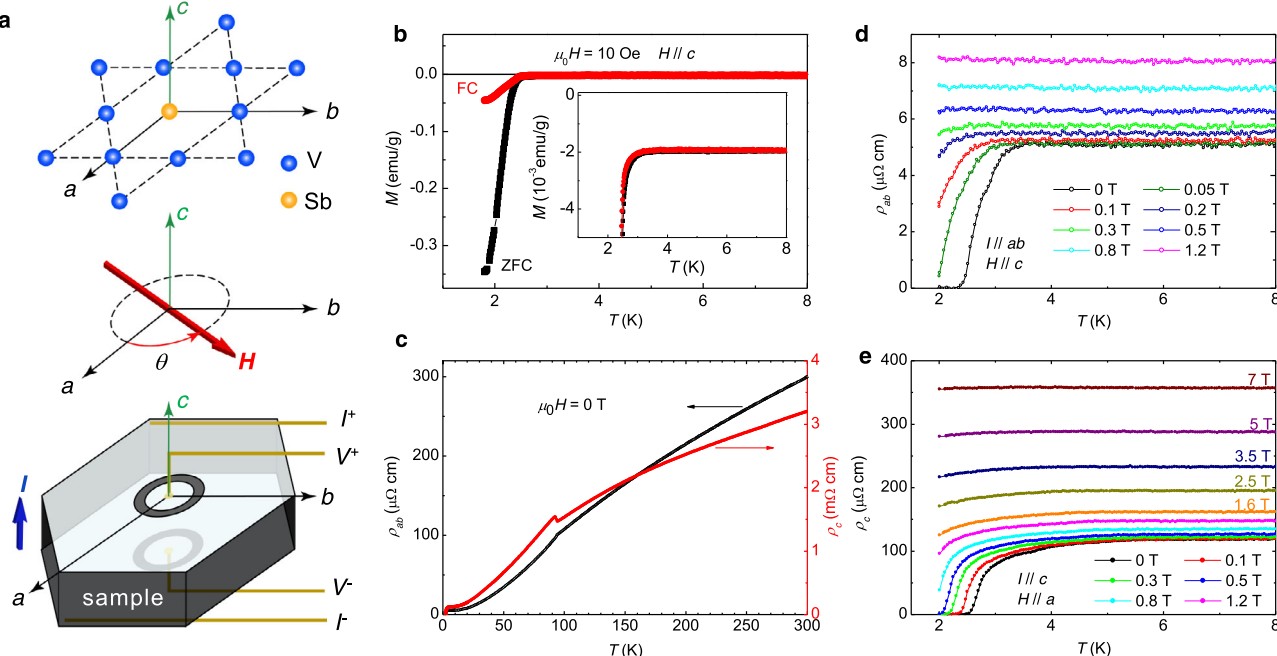

**Fig. 1 Measurement configuration of *c*-axis resistivity and characterization on superconductivity. a** The *c*-axis resistivity of CsV$_3$Sb$_5$ is measured by using a Corbino-shape-like electrode configuration. The electric current is applied mainly along *c*-axis of the single crystal, and the magnetic field is applied parallel to and rotated in the *ab*-plane. Single crystals usually have naturally formed edges with the angle of about 120° for neighboured edges, and these edges are along directions of crystallographic axes (see Supplementary Note 2). **b** Temperature-dependent magnetization measured with the zero-field-cooling (ZFC) and the field-cooling (FC) modes. **c** Temperature dependence of *c*-axis and in-plane resistivity which are measured with different configurations (Supplementary Fig. 3). Temperature dependence of in-plane (**d**) and *c*-axis (**e**) resistivity measured near the superconducting transition temperature at different magnetic fields.

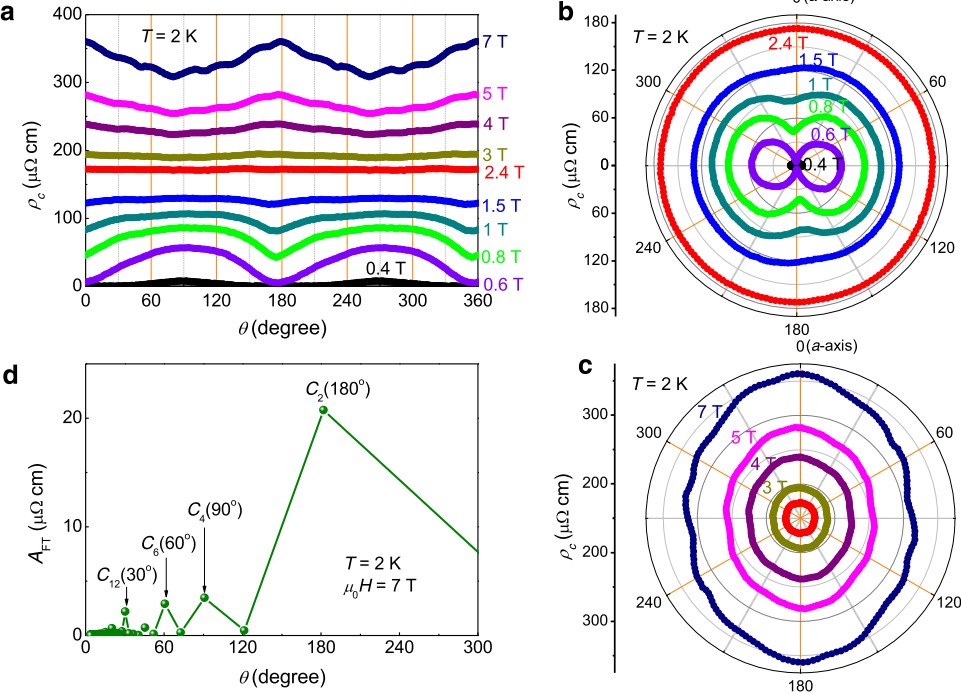

**Fig. 2 Twofold symmetry of angular dependent *c*-axis resistivity under in-plane magnetic field. a** Angular dependence of *c*-axis resistivity measured at different in-plane magnetic fields. The featureless $\rho_c(\theta)$ curve measured at 2.4 T separates two sets of $\rho_c(\theta)$ curves holding phase-reversed oscillations with twofold symmetry. Angular dependent *c*-axis resistivity plotted in polar coordinate measured with rotating in-plane magnetic field of the magnitude (**b**) below and (**c**) above 2.4 T. Local minima in curves measured at $\mu_0H < 2.4$ T change to local maxima in curves at $\mu_0H > 2.4$ T in the direction along *a*-axis. In addition, $\rho_c(\theta)$ curves measured at $\mu_0H > 2.4$ T show extra oscillation of six-fold symmetry besides the major twofold signal. **d** FT result to the $\rho_c(\theta)$ curve measured at 7 T.

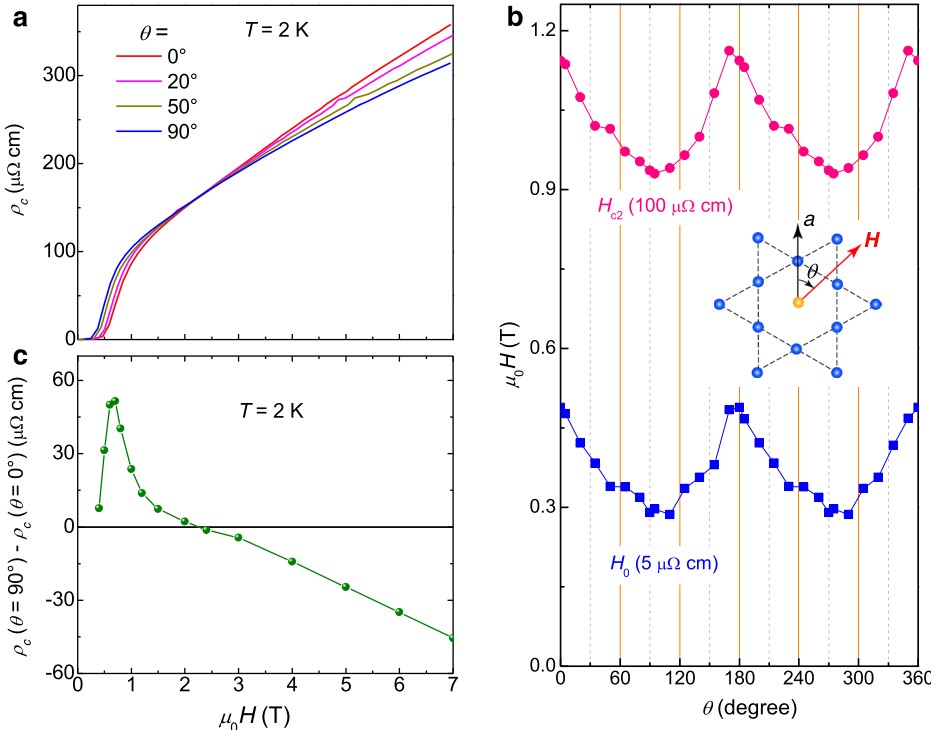

**Fig. 3 Magnetic-field induced phase reverse of $\rho_c(\theta)$ curves with twofold symmetry. a** Magnetic field dependence of $c$-axis resistivity measured with different angles between field and the $a$-axis at $T = 2$ K. **b** Angular dependent upper critical field and zero-resistance field ($\mu_0 H_0$) by using different criterions of $c$-axis resistivity. **c** Field dependence of the $c$-axis-resistivity difference between $\theta = 0°$ and $90°$. The resistivity difference changes its sign at a field of about 2.4 T.

stronger than 2.5 T at 2 K (Fig. 1e), the suppression of resistivity due to this effect is very weak when compared to the magnetoresistance. Thus the twofold symmetry of $\rho_c(\theta)$ curves at $\mu_0 H > 2.4$ T should be dominated by normal-state properties. In addition, some extra oscillations can be seen in $\rho_c(\theta)$ curves measured at very high field (Fig. 2a, c), see the curve measured at 7 T for example. Figure 2d shows the result of the Fourier transformation (FT) to the $\rho_c(\theta)$ curve measured at 7 T, and the FT amplitude ($A_{FT}$) peak at 180° (with the $C_2$ symmetry) combined with that at 90° ($C_4$) suggests the nematic electronic state, and the peaks at 60° ($C_6$) and 30° ($C_{12}$) suggest other six-fold symmetries. Actually, the $\rho_c(\theta)$ oscillates by every 30°, which is strongly correlated with the arrangement of vanadium atoms in the lattice[17] (detailed analysis see Supplementary Note 3).

The two kinds of $\rho_c(\theta)$ curves below and above 2.4 T are roughly orthogonal to each other in terms of the field direction of the extremum resistivity. This tendency can be clearly seen from the field-driven evolution of the angle-dependent $\rho_c(\mu_0 H)$ curves (Fig. 3a). By using certain criterions, namely $\rho_c(\mu_0 H) = 100$ and 5 μΩ·cm, we determined the angle-dependent upper critical field $\mu_0 H_{c2}$ and zero-resistance field $\mu_0 H_0$ (Fig. 3b). Now the peak position corresponds to the minimum resistivity in the superconducting state. The in-plane anisotropy of $\mu_0 H_{c2}$ is about 1.2−1.3, which is consistent with the value determined from temperature-dependent resistivity measurements (Supplementary Fig. 7d). The difference between the resistivity measured along two typical directions ($\theta = 0°$ and 90°) at 2 K is plotted in Fig. 3c with the variation of field. One can see that there is a clear sign change at the field of about 2.4 T. However, when $T = 10$ K, we see no cross of $\rho_c(\mu_0 H)$ curves (Supplementary Fig. 7e). Now if we take the characteristic fields $\mu_0 H^*$ with the criterions of $\rho_c(\mu_0 H^*) = 300$ or 320 μΩ·cm, we see again the twofold feature of $\mu_0 H^*$ (Supplementary Fig. 7f), but now $\mu_0 H^*(\theta)$ shows oscillations with opposite phase as that of $\mu_0 H_{c2}(\theta)$.

**Temperature evolution of twofold feature**. Figure 4a, b shows the temperature evolution of $\rho_c(\theta)$ curves at two different fields of 0.4 and 5 T, respectively. Obviously, the twofold feature of the $\rho_c(\theta)$ curve at 0.4 T weakens quickly with the increase of temperature. When $T$ reaches about 4 K, this oscillation is greatly diminished. This indicates that the twofold symmetry of $\rho_c(\theta)$ is just induced by the flux flow dissipation in the superconducting state. However, at roughly the same angles for the minimum resistivity in the superconducting state, the resistivity peaks up when $\mu_0 H = 5$ T (Fig. 4b). Figure 4c shows the difference of $\rho_c(\theta = 90°) − \rho_c(\theta = 0°)$ as an indicator to show the nematicity. Here the filled circles and squares represent the data for 0.4 and 5 T, respectively. With the increase of temperature, the twofold anisotropy of $\rho_c(\theta)$ curves measured at 0.4 T quickly vanishes around $T_c$, but that measured at 5 T changes much gently with the increase of temperature. We carry out the FT to $\rho_c(\theta)$ curves measured at different temperatures under 5 T, and the result is shown in Supplementary Fig. 8a. The FT amplitudes for the 180° peak are also plotted in Fig. 4c as another indicator to show the nematicity. One can see clearly that the twofold symmetry at 5 T progressively weakens with the increase of temperature and disappears near the CDW transition temperature ($T_{CDW}$) of about 95 K. In Fig. 5, we show the control experimental results carried out in another sample. One can also see the twofold symmetry of $\rho_c(\theta)$. The temperature evolution of the twofold feature is almost the same as the results shown in Fig. 4. The observations are very similar in these two samples.

## Discussion

Now we discuss the origin of the twofold symmetry observed in $\rho_c(\theta)$ curves. One may argue that this can be induced by the misalignment between the current direction and $c$-axis or that between the field and the $ab$-plane. Indeed, although we cannot

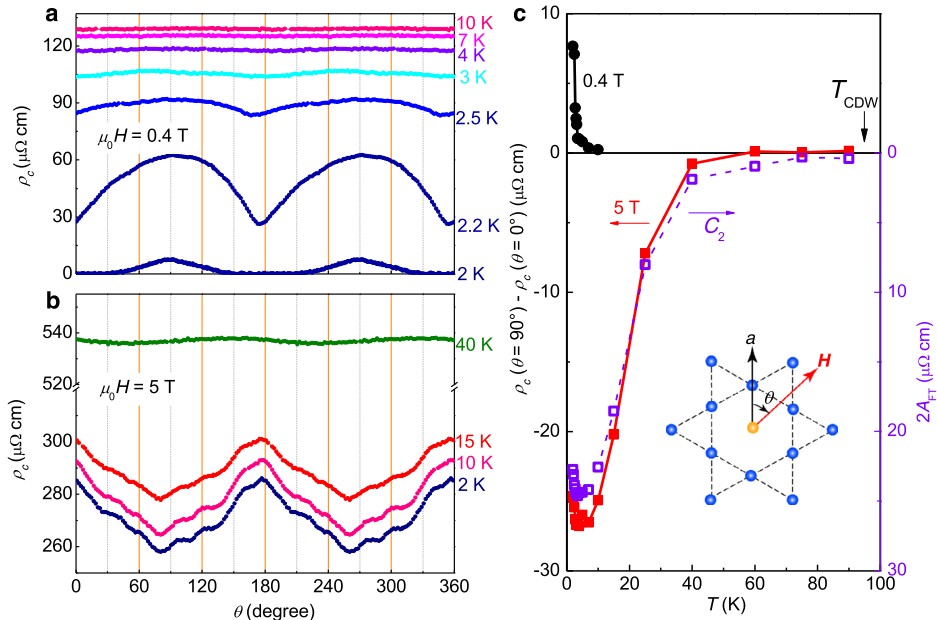

**Fig. 4 Temperature evolution of the nematicity of *c*-axis resistivity.** Angular dependent *c*-axis resistivity measured at different temperatures under a magnetic field of **a** 0.4 and **b** 5 T. **c** Temperature dependence of the nematicity. The solid symbols represent the *c*-axis resistivity difference between $\theta = 0°$ and 90° at 0.4 and 5 T, while the open symbols show twice the amplitude of the 180° (or $C_2$) peak ($2A_{FT,C2}$) from FT results (Supplementary Fig. 8a) to $\rho_c(\theta)$ curves measured at different temperatures under 5 T. One can see that the latter is more sensitive to show the $C_2$ symmetry which seems ending near $T_{CDW}$.

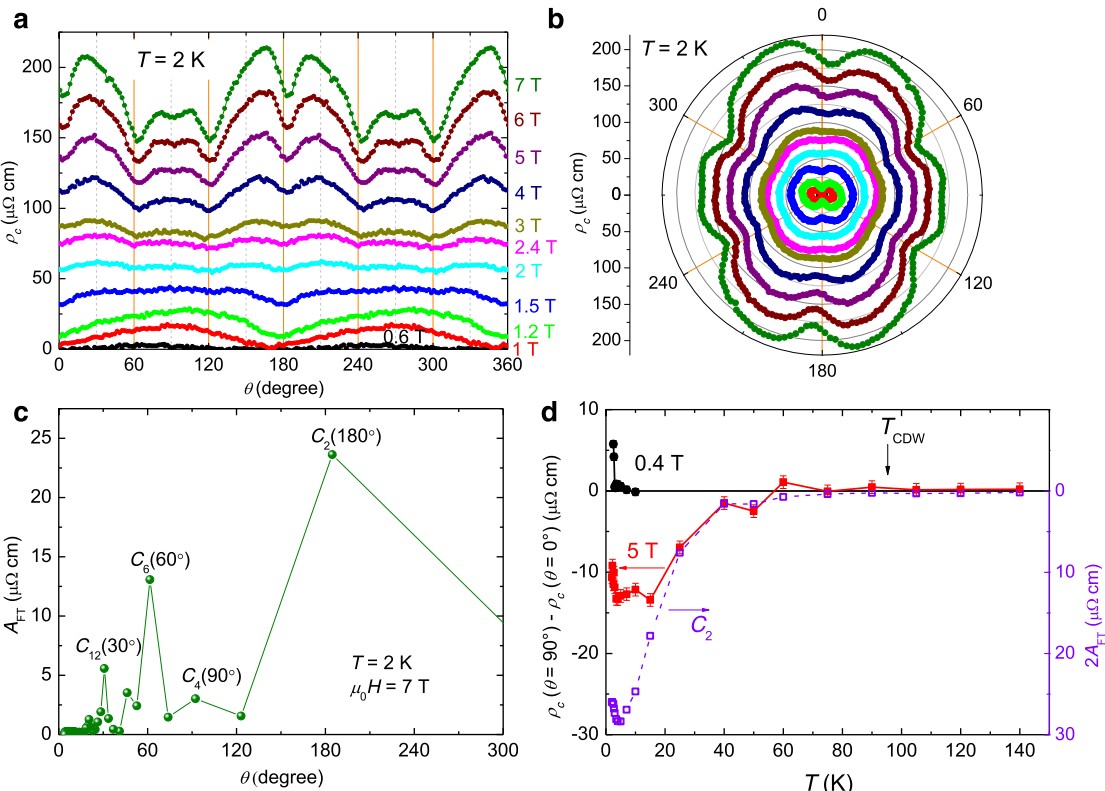

**Fig. 5 Control experiment of $\rho_c(\theta)$ curves with twofold symmetry of in another sample. a,b** Angular dependence of *c*-axis resistivity measured at different in-plane magnetic field shown by a rectangular and a polar coordinate, respectively. The inverse of the local extrema can be observed when the magnetic field crosses about 2 T. **c** FT result to the $\rho_c(\theta)$ curve measured at 7 T. **d** Temperature dependence of the nematicity. The solid symbols are derived from the *c*-axis resistivity difference between $\theta = 0°$ and 90° at 0.4 and 5 T, while the open symbols show twice the amplitude of the 180° (or $C_2$) peak ($2A_{FT,C2}$) from FT results (Supplementary Fig. 8c) to $\rho_c(\theta)$ curves measured at different temperatures under 5 T. The error bars in the figure are determined by the noise in the resistivity measurement and the error transfer theory. The relatively small thickness and large surface area make the resistance very small with a relatively large noise.

avoid this misalignment, however, this is unlikely for our results because of the following reasons. Firstly, for the $\rho_c(\theta)$ curve measured at 7 T and 2 K, $\rho_c(\theta = 90°)/\rho_c(\theta = 0°) = 0.85$ suggests a big difference. If the anisotropic magnetoresistance were induced by the misalignment between the magnetic field and the $ab$-plane, and the magnetoresistance was only induced by the $ab$-plane component of field which is perpendicular to the current, this anisotropy would correspond to a misalignment angle of about 32° (cos 32° = 0.85) between the field and the $ab$-plane. In the experiment, we can guarantee that the misalignment angle of the $ab$-plane to the field is less than 3°, thus it is impossible for such a big angle misalignment. Secondly, a relatively large normal-state resistance at 0° and 180° in the normal state would mean a large component of field perpendicular to the current (since we have a positive magnetoresistance), which would suppress super-conductivity more severely and also induce a larger flux-flow resistivity at the same angle. But this contradicts the observations. Thirdly, we have repeated the experiments in the same sample and in the re-cleaved sample (Supplementary Fig. 9), and we have also carried out the experiments in another sample (Fig. 5). These control experiments all show similar behaviours, indicating a high reproducibility. Although the directions of superconducting gap maximum are all along one pair of crystalline axes, these two directions have an intersect angle of about 60° (Supplementary Fig. 3) for the two samples which are mounted on the same sample holder. This excludes the possible error from the experimental setup.

In iron-based superconductors, the normal-state nematic electronic order was detected by measuring in-plane resistance with a detwinning setup at zero field[52]. However, this kind of in-plane resistive measurements for checking the $C_2$ symmetry cannot be easily done in the present $CsV_3Sb_5$ system. The reason is that the crystal structure has a fourfold symmetry in iron-based superconductors, and the sample shape is usually rectangular. Therefore, it is easy to measure the resistivity in $a$- or $b$-axis direction by using the standard four-probe technique on one sample, and the fourfold symmetry breaking can be easily obtained by the analysis of resistivity. However, in $CsV_3Sb_5$, the crystal structure has a six-fold symmetry, and the shape of the sample is usually hexagonal, with some corners of enclosed angles of about 120°. Hence, it is difficult to measure the exact resistivity along three crystalline directions in one sample mounted in a detwinning setup. Here, we detect the in-plane symmetry of the sample by measuring the $c$-axis resistivity with a rotating in-plane field, and the twofold symmetry in $\rho_c(\theta)$ curves may not reflect directly the nematic electron state in the $ab$ plane. However, the twofold symmetry nature of $\rho_c(\theta)$ curves in the normal state only show up when the magnetic field is higher than 2.4 T at 2 K. In the presence of a magnetic field, the mobile electrons will possess a circular momentum in the plane perpendicular to the field direction. Then the $c$-axis resistivity measured in this configura-tion should contain the contribution of the in-plane electronic states or the in-plane mobility component along the direction perpendicular to the magnetic field[66]. For example, if the in-plane Fermi velocity has a two-fold symmetry, that will induce a two-fold symmetric feature of the in-plane scattering rate, which can be detected by the $c$-axis resistivity with an in-plane magnetic field.

The twofold symmetry of $\rho_c(\theta)$ curves measured at high fields may suggest the in-plane nematic electronic state in the material. Based on the FT results obtained at high temperatures across $T_{CDW}$ (Supplementary Fig. 8d, f), one can see that the amplitude of the 180° (or $C_2$) peak is reduced to the background signal when the temperature is increased to that near $T_{CDW}$, and the ampli-tude is always in the noise level at high temperatures above $T_{CDW}$. Consequently, the first scenario comes to our mind is that it may

be related to the CDW state. We have been aware that there is an additional $4a_0$ unidirectional charge order besides the tri-directional charge order with a $2a_0$ period in this material[24,42,44]. This unidirectional charge order pattern corre-sponds to CDW stripes along $a$-axis, which may be explained based on the picture of topological CDW[67]. These CDW stripes induce a twofold symmetry of in-plane electronic properties, which has been detected by our $c$-axis resistivity when the mag-netic field is rotated within the V−Sb planes. In addition, for the tri-directional charge order with a $2a_0$ period, the intensities of three sets of peaks have pronounced intensity anisotropy[24,41–44], which exactly shows the feature as the nematic CDW state. Besides, theoretically, it was predicted that a chiral flux phase may exist and break the time-reversal symmetry in the $c$-axis[68], but this model cannot predict a symmetry breaking in the $ab$ plane. Although the six-fold rotational symmetry is supposed to be preserved in a single V−Sb kagome layer, the stacking of these layers along the $c$-axis may induce the $C_6$ symmetry breaking. The three-dimensional (3D) $2a_0 \times 2a_0 \times 2c_0$ CDW order[25,34] or $2a_0 \times 2a_0 \times 4c_0$ one[36] was identified by different kinds of experiments. In this 3D CDW configuration, there is a π phase shift between neighboured V−Sb kagome layers, i.e., a misalignment of $a_0$ emerges along one of three in-plane crystalline axes for the same characteristic in-plane CDW patterns in neighboured layers. Therefore, the phase shift in addition to the inter-layer coupling between the neighboured layers lower the six-fold symmetry to a twofold one[39,40]. This picture does not require any additional type of order, and it can naturally explain the twofold symmetry of $\rho_c(\theta)$ curves with an in-plane rotating magnetic field in the normal state. A simple understanding is that the 3D CDW phase affects the in-plane mobility or the scattering rate along and perpendicular to the symmetry breaking axis. It should be noted that the $\rho_c(\theta,B)$ curve at a fixed angle $\theta$ should be an even function of the magnetic field[66], then the calculated resistivity difference $\rho_c(\theta = 90°) − \rho_c(\theta = 0°)$ is likely to be proportional to $B^2$ in the low field region. This fact is demonstrated in Supplementary Fig. 10, the nematicity effect appears with the presence of a small field and it is enlarged by a high magnetic field.

The twofold feature of $\rho_c(\theta)$ curve in the superconducting state may be intimately related to the feature in the normal state. A simple picture is that the CDW phase with twofold symmetry would gap out the density of states at the Fermi level leading to a truncated Fermi surface with twofold symmetry, and this leads to a twofold symmetry of the superconducting gap or $H_{c2}$. By now the existence of the gap anisotropy is still hard to be detected directly from experiments[25]. Alternatively, concerning the fact that the nematic electronic state is observed with the help of a strong in-plane magnetic field with the absence of super-conductivity, while the twofold symmetry of $H_{c2}(\theta)$ is observed in superconducting transition region with very small field: these two kinds of twofold symmetries may have different origins. We have noticed that the material is supposed or partly proved to be a topological superconductor[17,25,45,46,48], thus the super-conductivity with twofold symmetry may originate from the superconducting order parameter with odd parity. This scenario has recently been well proved[56–61,64,65] in topological super-conductors $Cu_xBi_2Se_3$, $Sr_xBi_2Se_3$, or $Nb_xBi_2Se_3$. Beside these two possibilities, a $4a_0/3$ bidirectional pair density wave[24] or the spin-triplet superconductivity[27] may be extra possible reasons of the twofold symmetry of $\rho_c(\theta)$ curves at small magnetic field in the superconducting state, thus quantitative analyses based on these models are highly desired.

As mentioned above, the natural explanation for the twofold symmetry of $\rho_c(\theta)$ curves is the 3D CDW configuration, and the π phase shift between neighboured V−Sb kagome layers emerges along one of three in-plane crystalline axes. Based on this picture,

there may be three kinds of nematic domains, and the angle between every two kinds of domains is 60°. The twinned $C_2$ electronic state may exist in the samples, and the existence of multiple nematic domains may be supported by the angle deviation between twofold and six-fold components derived from $\rho_c(\theta)$ curves (Supplementary Fig. 6). If there were quite a lot of randomly distributed domains, the twofold symmetric $\rho_c(\theta)$ curves would not be observed. However, we do observe the twofold nature of $\rho_c(\theta)$ curves both in superconducting and normal states. In addition, these two kinds of orders are orthogonal to each other in terms of the field direction of the minimum resistivity. Nevertheless, one kind of the nematic domain dominates the electronic transport as seen from the $\rho_c(\theta)$ curves. It should be noted that the nematic superconductivity is also observed in some topological superconductors by different kinds of experiments without the detwinning operation[56–60], and the multi-domain effect cannot induce the equal concentrations of different domains. The nematic electronic state is also observed in ultra-clean quantum Hall systems[54] and $Sr_3Ru_2O_7$[55] without detwinning. In our point of view, the situation in $CsV_3Sb_5$ is similar to that in the materials referenced above. Based on our experimental data, we argue that the predominant contribution of one kind of domain may be fixed by the subtle stress in the sample due to the different thermal shrinkage coefficients of the sample and the grease used to attach the sample on the sample holder. With the increase of temperature, although the twofold symmetry of $\rho_c(\theta)$ curves remains, the phase of the twofold component changes in all the samples (Supplementary Figs. 11−13). This means that the contribution from other domains increases. That may be the reason why the anisotropy disappears at about 60 K if we use $\rho_c(\theta = 90°) - \rho_c(\theta = 0°)$ as the indicator of the nematicity. However, the twofold symmetry feature does disappear near $T_{CDW}$ as seen from the measured $\rho_c(\theta)$ curves (Supplementary Figs. 11−13) or the temperature-dependent FT amplitude of the $C_2$ peak (Figs. 4c, 5d, and Supplementary Fig. 13l), which confirms a very close relationship between the twofold symmetry of $\rho_c(\theta)$ curves and 3D CDW transition.

In summary, we have observed twofold symmetry of superconductivity and the anti-phase oscillation of $c$-axis resistivity in the normal state with respect to the in-plane magnetic field, and these observations will shed new light in the study of this fascinating kagome and topological material.

## Methods

**Single-crystal growth and preparation**. High-quality single crystals of $CsV_3Sb_5$ were grown by a self-flux method with Cs−Sb binary eutectic mixture as the flux[16,44]. High-purity elements of Cs, V, and Sb were mixed in a molar ratio of 7:3:14 in an alumina crucible, and then the alumina crucible was sealed inside an evacuated quartz tube. The loaded tube was put into a muffle furnace and heated slowly to 1000 °C, and this temperature was kept for 20 h. After that, the tube was cooled down to 200 °C at a rate of 3 °C/h. Flake-like single-crystalline samples have the $c$-axis orientation. The analysis of the structure and the stoichiometry to $CsV_3Sb_5$ single crystals can be seen in Supplementary Note 1. The crystal orientation was determined by the Laue X-ray crystal alignment system (Photonic Science Ltd.). Some single crystals have naturally formed edges with the angle of about 120° for neighboured edges, and these edges are proved to be crystallographic axes by Laue diffraction measurements (see Supplementary Note 2).

**Resistivity measurements**. Resistance measurements were carried out in a physical property measurement system (PPMS, Quantum Design). Samples were cleaved along the Van der Waals layers, and some edge(s) were cut in order to form the hexagon structure by following the naturally formed edges (see Supplementary Fig. 3). The $c$-axis resistance was measured by the four-electrode method with the Corbino-shape-like configuration[60]. To eliminate the influence of the slight Hall signals on the raw data of angular dependence of resistivity, the resistivity taken at every angle has been averaged with positive and negative magnetic fields. The in-plane resistivity $\rho_{ab}$ was measured by the standard four-electrode method with remade electrodes on the same sample, and the current was in the $ab$-plane of the sample.

## Data availability

All the data that support the findings of this paper are available from the corresponding authors upon reasonable request.

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

## Acknowledgements

We acknowledge helpful discussions with Yaomin Dai. We appreciate the kind help in the analysis of the crystal alignment data given by Ning Yuan. This work was supported by National Natural Science Foundation of China (Nos. 11927809, 11974171, 12061131001, 92065109, 11734003, and 1904294), the National Key R&D Programme of China (No. 2020YFA0308800), Strategic Priority Research Programme (B) of the Chinese Academy of Sciences (No. XDB25000000), Beijing Natural Science Foundation (No. Z190006), and Beijing Institute of Technology Research Fund Programme for Young Scholars (No. 3180012222011).

## Author contributions

Y.L., Z.W, and Y.Y. grew single crystals and characterized the structure and stoichiometry of samples. Q.L. and W.X. measured and analyzed the crystal orientation. Y.X., H.Y., and H.-H.W. carried out resistivity measurements. H.Y., H.-H. W., Y.X., and Q.L. analyzed the data and wrote the manuscript which was proof-read and agreed by all authors.

## Competing interests

The authors declare no competing interests.
