## [Peer Review File · Nature Communications]

REVIEWER COMMENTS

Reviewer #1 (Remarks to the Author):

This is a timing experimental work on a newly discovered kagome superconductor AV₃Sb₅. The authors find that the c-axis resistivity shows an unusual twofold symmetry with in-plane magnetic field. The field and temperature dependent resistivity measurements are interesting and possibly related to the novel behaviors observed in STM studies. However, there are several critical issues that prevent me to recommend the current manuscript to publish in Nature Communications.

1. I'd like to remind the authors that the 2*2*2 (or 2*2*4) CDW/structural transition near 90 K breaks the C₆ symmetry. I'm not surprised that the resistivity follows the C₂ symmetry.

2. Assuming an additional electronic C₂ symmetry, I would expect a twinned C₂ electronic state. It is not clear to me why the C₂ symmetry is observed.

3. The discussion is too speculative. The authors provide insightful discussions and connect their observations with previous results. However to publish in high-impact journals, I would expect a convincing and new result that deepens our understanding of this new material.

Reviewer #2 (Remarks to the Author):

The manuscript by Xiang et al concerning the anisotropic magnetogalvanic charge transport provides some compelling new data on an interesting class of materials. This paper wisely does not seek to claim they have solved all of the exotic charge phenomena in these materials but instead appears to provide one important dataset. Further details concerning crystal growth and ultimate stoichiometry could be provided, as the quality/stoichiometry of the samples (particularly with respect to cation vacancies) has previously been found to be important.

Reviewer #3 (Remarks to the Author):

The authors reported the twofold symmetry of c-axis resistivity with in-plane rotation of magnetic field in the recently discovered kagome metal CsV₃Sb₅. In the superconducting phase, the twofold symmetry of c-axis resistivity appeared at low in-plane magnetic field. But, in the normal state, such anisotropy can only appear at relatively high in-plane magnetic field. Currently, the kagome metals are under extensive study due to their unconventional correlated behavior. This study is timely and interesting but, as a theorist, I am still not persuaded that the observed c-axis resistivity is related to the intrinsic nematic order. The manuscript needs to clarify the following points,

1.The nematic order in iron base superconductors emerges spontaneously. Here, however, the anisotropic c-axis resistivity emerges with an in-plane magnetic field. How to rule out the anisotropy in the superconducting phase originated from the in-plane magnetic field (even though the magnetic field is small)?

2.In the normal states, the nematic order is usually coupled with lattice distortions and the anisotropy can persist above the structural phase transition. In present paper, this anisotropy can only be observed with a high magnetic field and the authors argued that the nematic order may be related to the CDW order. According to the data, the anisotropy disappears at 50 K (H=5 T), which is much lower than the CDW phase transition temperature (about 90 K). Do the authors have a good understanding for this? Why the anisotropy is not observed in the normal state with a low in-plane magnetic field (between 20 and 50 K) if a nematic order emerges in the CDW phase?

3.As the kagome metals are hexagonal, the period of 60 appears in the Fourier transformation for c-axis resistivity. In Fig.2(d) and Fig.4(c), why there is no obvious peak at 120 degree but a noticeable peak at 90 degree? Will this peak at 120 degree appear when the nematic order vanishes?

Responses to the Reviewers' report

Responses to the Report of Reviewer #1

This is a timing experimental work on a newly discovered kagome superconductor AV₃Sb₅. The authors find that the c-axis resistivity shows an unusual twofold symmetry with in-plane magnetic field. The field and temperature dependent resistivity measurements are interesting and possibly related to the novel behaviors observed in STM studies. However, there are several critical issues that prevent me to recommend the current manuscript to publish in Nature Communications.

Response: We thank the reviewer for the appreciation of our work. The reviewer raised several constructive suggestions, and we have modified our manuscript according to these suggestions/questions. We hope our responses can remove the concerns of this reviewer.

1. I'd like to remind the authors that the 2*2*2 (or 2*2*4) CDW/structural transition near 90 K breaks the C₆ symmetry. I'm not surprised that the resistivity follows the C₂ symmetry.

Response: We thank the reviewer for reminding us of this issue. We agree with the reviewer that the 2x2x2 (or 2x2x4) CDW/structural transition does break the C₆ symmetry and that could be the right reason for the in-plane twofold electronic properties observed by us. However, we recall to the reviewer that our original manuscript reported a very early work showing the experimental findings of the anisotropic in-plane electronic state as well as twofold symmetry of superconductivity by bulk transport measurements in this kagome material. At that time, the STM works had not disclosed the anisotropic intensities of the FT-QPI patterns, and an explicit theoretical picture was lacking to explain the origin of our observations. With time elapsing, more experimental and theoretical works help us to further understand the C₂ symmetry of the material.

In the revised version, we add some discussions based on this picture, and we also add several recent works as references (Refs. 36, 39, and 40). These discussions read as "Although the six-fold rotational symmetry is supposed to be preserved in a single V-Sb kagome layer, the stacking of these layers along the c-axis may induce the C₆ symmetry breaking. The three-dimensional (3D) 2a₀x2a₀x2c₀ CDW order^{25,34} or 2a₀x2a₀x4c₀ one³⁶ was identified by different kinds of experiments. In this 3D CDW configuration, there is a π phase shift between neighbored V-Sb kagome layers, i.e., a misalignment of a₀ emerges along one of three in-plane crystalline axes for the same characteristic in-plane CDW patterns in neighbored layers. Therefore, the phase shift in addition to the inter-layer coupling between the neighbored layers lower the six-fold symmetry to a twofold one^{39,40}. This picture does not require any additional type of order, and it can naturally explain the twofold symmetry of $\rho_c(\theta)$ curves with an in-plane rotating magnetic field in the normal state."

2. Assuming an additional electronic C2 symmetry, I would expect a twinned C2 electronic state. It is not clear to me why the C2 symmetry is observed.

Response: This is actually a good question. The reviewer is right that the twinned C_2 electronic state may exist in the samples. The existence of multiple nematic domains may be supported by the angle deviation between twofold and six-fold components derived from $\rho_c(\theta)$ curves (Supplementary Fig. 6). As reminded by this reviewer, the phase shift between neighbored kagome layers and the inter-layer coupling can naturally induce the bulk six-fold symmetry breaking. Based on this picture, there may be three kinds of nematic domains, and the angle between every two kinds of domains is 60° . If there were quite a lot of randomly distributed domains, the twofold symmetric $\rho_c(\theta)$ curves would not be observed. However, we do observe the twofold nature of $\rho_c(\theta)$ curves both in superconducting and normal states. In addition, these two kinds of orders are orthogonal to each other in terms of the field direction of the minimum resistivity. It means that one kind of the nematic domain dominates the electronic transport for reasons as detailed below. It should be noted that the nematic superconductivity is also observed in topological superconductors by different kinds of experiments without the detwinning operation. The multi-domain effect is discussed in Refs. 55, and this effect cannot induce equal concentrations of different domains. The nematic electronic state is also observed in ultra-clean quantum Hall systems and $\text{Sr}_3\text{Ru}_2\text{O}_7$ without detwinning. In our point of view, the situation in CsV_3Sb_5 is similar to that in materials referenced above.

Based on our experimental data, we argue that the predominant contribution of one kind of domain may be fixed by the subtle stress in the sample due to the different thermal shrinkage coefficients of the sample and the grease used to attach the sample on the sample holder. With increase of temperature, although the twofold symmetry of $\rho_c(\theta)$ curve exists, the phase of the twofold component changes. This means that the contribution from other domains increases. In the revised version, we add some related discussions to the main text, and we also add Supplementary Figs. 11 and 12 to illustrate this issue. Based on the detailed analysis on the intensity of the Fourier transform spots, we also confirm that the twofold symmetry of $\rho_c(\theta)$ curve disappears near the CDW transition temperature instead of 60 K. We have added the related discussions to the last part of the Discussion section.

3. The discussion is too speculative. The authors provide insightful discussions and connect their observations with previous results. However to publish in high-impact journals, I would expect a convincing and new result that deepens our understanding of this new material.

Response: We thank the reviewer for the suggestion. In the revised version, we add new discussions of the possible origins of the nematic phase, the multi-domain effect, and the temperature and magnetic field dependent evolution of the twofold component. Now the

discussion part becomes much more fruitful. We hope this reviewer is satisfied with our revised version and the thoughtful explanations.

Responses to the Report of Reviewer #2

The manuscript by Xiang et al concerning the anisotropic magnetogalvanic charge transport provides some compelling new data on an interesting class of materials. This paper wisely does not seek to claim they have solved all of the exotic charge phenomena in these materials but instead appears to provide one important dataset. Further details concerning crystal growth and ultimate stoichiometry could be provided, as the quality/stoichiometry of the samples (particularly with respect to cation vacancies) has previously been found to be important.

Response: We thank this reviewer for the kind appreciation to our work. We also thank the reviewer for this very useful suggestion. In the revised version, we add the details of the crystal growth to Methods. We also add the XRD data and the EDS data to the Supplementary Materials as Supplementary Figs. 1 and 2, respectively. In addition, we add some discussions on the structure and stoichiometry of the single crystal as Supplementary Note 1. We hope this reviewer will be satisfied with these additional data and discussions.

Responses to the Report of Reviewer #3

The authors reported the twofold symmetry of c-axis resistivity with in-plane rotation of magnetic field in the recently discovered kagome metal CsV₃Sb₅. In the superconducting phase, the twofold symmetry of c-axis resistivity appeared at low in-plane magnetic field. But, in the normal state, such anisotropy can only appear at relatively high in-plane magnetic field. Currently, the kagome metals are under extensive study due to their unconventional correlated behavior. This study is timely and interesting but, as a theorist, I am still not persuaded that the observed c-axis resistivity is related to the intrinsic nematic order. The manuscript needs to clarify the following points,

Response: We appreciate the careful review and precise judgment to our manuscript by this reviewer. We also appreciate constructive suggestions raised by this reviewer. We have improved our manuscript according to these suggestions, and we hope the reviewer will be satisfied with the revised version.

1. The nematic order in iron base superconductors emerges spontaneously. Here, however, the anisotropic c-axis resistivity emerges with an in-plane magnetic field. How to

rule out the anisotropy in the superconducting phase originated from the in-plane magnetic field (even though the magnetic field is small)?

Response: We thank the reviewer for the careful review. Indeed, in iron-based superconductors, the nematic electronic order emerges spontaneously in normal state, and is detected by measuring in-plane resistance in a detwinning setup at zero field. We believe the in-plane C_2 symmetric properties in CsV_3Sb_5 also emerge simultaneously, perhaps together with, or slightly below the CDW transition temperature. As we added a new possible origin for explaining twofold feature, namely the $2\times 2\times 2$ CDW order, it has a natural C_2 symmetry. However, we want to note that this kind of in-plane resistive measurements for checking the C_2 symmetry cannot be easily done in the present CsV_3Sb_5 system as in iron-based superconductors. The reason is that, the crystal structure has a fourfold symmetry in iron-based superconductors, and the sample shape is usually rectangular. Therefore, it is easy to measure the resistivity in a - or b -axis direction by using the standard four probe technique on one sample, and the fourfold symmetry breaking can be easily obtained by the analysis of the resistivity. However, in CsV_3Sb_5 , the crystal structure has a six-fold symmetry, and the shape of the sample is usually hexagonal, with some corners of enclosed angles of about 120° . So it is difficult to measure the exact resistivity along three crystalline directions in one sample mounted in a detwinning setup. In this manuscript, we detect the in-plane symmetry of the sample by measuring the c -axis resistivity at a rotating in-plane field. In the presence of a high magnetic field, the mobile normal-state electrons will possess a circular momentum in the plane perpendicular to the field direction. Thus the c -axis resistivity measured in this configuration should be able to detect the contribution from the in-plane electronic states along the direction perpendicular to the magnetic field in the normal state. The nematic in-plane electronic state is likely to be induced by the three-dimensional $2\times 2\times 2$ or $2\times 2\times 4$ CDW order as mentioned by the first reviewer. The phase shift between kagome layers and the inter-layer coupling can naturally induce the bulk six-fold symmetry breaking, leading to a C_2 symmetry. We have added some discussions to the main text in the revised version.

Concerning whether “the anisotropy in the superconducting phase is originated from the in-plane magnetic field”, we thus intentionally measure the resistivity along c -axis and with a rotating in-plane magnetic field. In this configuration, the averaged current direction (along c -axis) is always perpendicular to the in-plane magnetic field, this greatly lowers down the possibility that the flux flow resistivity has an artificial angle dependence if the current were applied exactly in the ab plane. We assume that the reviewer may worry about the imperfect alignment of the in-plane magnetic field, thus there is a c -axis component of magnetic field and the effective component of the in-plane magnetic field gives rise to a fake twofold symmetry. But we believe this cannot be the major reason of the twofold symmetry of superconductivity, even there is a tiny misalignment. In our experiment, we can guarantee that the misalignment angle of the ab -plane to the field is less than 3° . Then the c -axis component is about 5% of the applied magnetic field. At 2.2 K, the upper critical field is about 2.38 T for $H//a$ and 1.85 T for $H\perp a$ with the criterions of $90\%\rho_n$. If we use 5% to calculate the c -axis component at about 2 T, that is about 0.1 T.

However, the upper critical field is about 0.15 T for $H//c$ at 2.2 K from our resistivity measurements. In this point of view, the c -axis component of magnetic field, even existing due to the maximum misalignment, is not strong enough to induce the angle dependent difference. The other two reasons to rule out this possibility have been proposed in the main text, we repeat them here. “Secondly, a relatively large normal-state resistance at 0° and 180° in the normal state would mean a large component of field perpendicular to the current (since we have a positive magnetoresistance), which would suppress superconductivity more severely and also induce a larger flux-flow resistivity at the same angle. But this contradicts the observations. Thirdly, we have repeated the experiments in the same sample (Supplementary Fig. 9) and another sample (Fig. 5) as control experiments, and all show the same behaviours, indicating a high reproducibility. Although the directions of superconducting gap maximum are all along one pair of crystalline axes, these two directions have an intersect angle of about 60° (Supplementary Fig. 3) for the two samples which are mounted on the same sample holder.” Based on the three reasons mentioned above, we can exclude the possibility that the twofold symmetry of superconductivity is induced by a small c -axis component of the magnetic field due to misalignment.

2. In the normal states, the nematic order is usually coupled with lattice distortions and the anisotropy can persist above the structural phase transition. In present paper, this anisotropy can only be observed with a high magnetic field and the authors argued that the nematic order may be related to the CDW order. According to the data, the anisotropy disappears at 50 K ($H=5$ T), which is much lower than the CDW phase transition temperature (about 90 K). Do the authors have a good understanding for this?

Response: We thank this reviewer for the careful review and this comment. In the original version, we use the resistivity difference measured at $\theta = 0^\circ$ and 90° as the indicator to show the nematicity. However, inspired by this reviewer and the first reviewer, this indicator is not suitable for the multi-domain situation. Based on the picture of the 3D CDW, there exist three kinds of nematic domains, and the angle between every two kinds of domains is 60° . At low temperatures, we do observe the twofold nature of $\rho_c(\theta)$ curves both in superconducting and normal states. In addition, these two kinds of orders are orthogonal to each other in terms of the field direction of the minimum resistivity. It means that one kind of the nematic domain dominates the electronic transport. Based on our experimental data, we argue that the predominance of one kind of domain may be fixed by the subtle stress in the sample due to the different thermal shrinkage coefficients of the sample and the grease used to attach the sample on the sample holder. However, the situation changes with increase of temperature. At high temperatures, although the twofold symmetry of $\rho_c(\theta)$ curve exists, the phase of the twofold component changes. This means that the relative contribution from other domains increases. In the newly added Supplementary Figs. 11 and 12, we show the $\rho_c(\theta)$ curves measured at different temperatures. One can see clearly that the phase of the twofold component shifts with increase of temperature. That may be the reason why the anisotropy disappears at about

50 K if we use $\rho_c(\theta = 90^\circ) - \rho_c(\theta = 0^\circ)$ as the indicator of the nematicity. However, the twofold symmetry feature disappears near the CDW transition temperature from the measured $\rho_c(\theta)$ curves. In the revised version, we add the Fourier transform amplitude of the twofold component as a more accurate index of the nematicity. Based on the newly added data, we can conclude that the nematic electronic state is closely related to the CDW transition. We also add some discussions to the related part to address this issue.

Why the anisotropy is not observed in the normal state with a low in-plane magnetic field (between 20 and 50 K) if a nematic order emerges in the CDW phase?

Response: We thank this reviewer for the careful review. The reviewer is right that the observed magnitude of nematicity seems to be enhanced with increase of magnetic field because here we use the magnetoresistance as the indicator. In Supplementary Fig. 10, we show this effect by calculating the magnetic field dependence of $\rho_c(\theta = 90^\circ) - \rho_c(\theta = 0^\circ)$. We should emphasize that the nematicity is in the *ab* plane, but what we measured is the *c*-axis resistivity. Obviously, the nematicity cannot be detected at 0 T. In the presence of magnetic field, the mobile electrons will possess a circular momentum in the plane perpendicular to the field direction. Then the *c*-axis resistivity measured in this configuration should contain the contribution from the in-plane electronic states or the in-plane mobility component along the direction perpendicular to the magnetic field. In the revised version, we add some discussions as “A simple understanding is that the 3D CDW phase affects the in-plane mobility or the scattering rate along and perpendicular to the symmetry breaking axis. It should be noted that the $\rho_c(\theta, B)$ curve at a fixed angle θ should be an even function of the magnetic field⁶⁵, and then the calculated resistivity difference $\rho_c(\theta = 90^\circ) - \rho_c(\theta = 0^\circ)$ is likely to be proportional to B^2 in the low field region. This fact is demonstrated in Supplementary Fig. 10; the nematicity effect appears with the presence of a small field and it is enlarged by a high magnetic field.”

3. As the kagome metals are hexagonal, the period of 60 appears in the Fourier transformation for *c*-axis resistivity. In Fig.2(d) and Fig.4(c), why there is no obvious peak at 120 degree but a noticeable peak at 90 degree? Will this peak at 120 degree appear when the nematic order vanishes?

Response: We thank this reviewer for this pertinent comment. In fact, the Fourier transformation (FT) to a real function is an algorithm by using a series of cosine functions to express this function, and the FT amplitude is the absolute value of the coefficient of the obtained cosine functions. These amplitudes are independent to each other. The reviewer is right that there is a period of 60° in the FT to the *c*-axis resistivity, and this angle corresponds to the six-fold symmetry. It should be noted that there is also a period of 30° in the FT to the *c*-axis resistivity, and this angle corresponds to a 12-fold symmetry. The 12-fold symmetric component is certainly six-fold symmetric, but the FT amplitude for the 12-fold symmetric component is independent of the six-fold symmetric component. In

Supplementary Fig. 6b,d, the six-fold component curve is the summation of six-fold (with the period of 60°) and 12-fold (with the period of 30°) symmetric components. Similarly, the 12-fold or six-fold symmetric component is certainly threefold symmetric, but the FT amplitude for the threefold symmetry is independent of that for the 12-fold or six-fold symmetry. Since the magnetoresistance should be an even function of the magnetic field and the in-plane structure is six-fold symmetric, the component with a period of 120° , which corresponds to a threefold symmetry with the mirror symmetry breaking, is generally very small.

Being similar to the six-fold and 12-fold symmetric component, the fourfold symmetric component is also twofold symmetric. This means that the nematicity cannot be easily expressed as a cosine function with period of 180° , and the fourfold symmetric component should be required to further refine the detail of the nematicity. We add the statement in the main text to emphasize the issue as “Figure 2d shows the Fourier transformation (FT) to the $\rho_c(\theta)$ curve measured at 7 T, and the FT amplitude (A_{FT}) peak at 180° (C_2) combined with that at 90° (C_4) suggests the nematic electronic state, and the peaks at 60° (C_6) and 30° (C_{12}) suggest other six-fold symmetries.” We also add some discussions to Supplementary Note 3 to clarify this point to the readers as “It should be noted that the fourfold symmetric component is also twofold symmetric, i.e., the fourfold symmetric component should have two mirror symmetrical axes. This means that the nematicity cannot be easily expressed as a cosine function with period of 180° , and the fourfold symmetric component is required to further refine the detail of the nematicity. Similarly, the 12-fold symmetric component is certainly six-fold symmetric. Therefore, we use the summation of components with twofold (with the period of 180°) and fourfold (with the period of 90°) symmetry as the contribution from the nematic electronic states, while we use the summation of the components with six-fold (with the period of 60°) and 12-fold (with the period of 30°) symmetry as the contribution from the electronic states due to the six-fold electronic structure in this material.”

For the six-fold symmetric component, the feature is rapidly suppressed by temperature. In the newly added Supplementary Figs. 8, 11, and 12, the six-fold feature cannot be clearly identified at about 40 K. We add this information to the legends of Supplementary Figs. 11 and 12.

REVIEWER COMMENTS

Reviewer #1 (Remarks to the Author):

The authors properly addressed my concerns, especially the importance of $2 \times 2 \times 2(4)$ CDW that breaks C_6 rotational symmetry. The discussion on domain effect is also satisfactory. In the revised manuscript, the authors provide additional evidence to link the CDW phase and anisotropic resistivity. I think this information is very important for a fast developing field and should be published in high visible journals.

Before I recommend it to publish in Nature Communications, there's one critical issue that need to be fixed. In figure 4c and 5d, only $T < T_{CDW}$ anisotropy data is presented. The authors should show at least two more data points for $T > T_{CDW}$ to reveal the possible connection between $2 \times 2 \times 2(4)$ CDW and anisotropic resistivity.

Reviewer #3 (Remarks to the Author):

The authors have properly addressed the points I previously raised. This observed interesting twofold symmetry of c-axis resistivity will stimulate further studies on the correlated phenomena in kagome metal CsV_3Sb_5 . I can agree with this paper being published in Nature Communications if other referees and the editors agree.

Responses to the Reviewers' report

Responses to the Report of Reviewer #1

The authors properly addressed my concerns, especially the importance of $2 \times 2 \times 2(4)$ CDW that breaks C_6 rotational symmetry. The discussion on domain effect is also satisfactory. In the revised manuscript, the authors provide additional evidence to link the CDW phase and anisotropic resistivity. I think this information is very important for a fast developing field and should be published in high visible journals.

Before I recommend it to publish in Nature Communications, there's one critical issue that need to be fixed. In figure 4c and 5d, only $T < T_{CDW}$ anisotropy data is presented. The authors should show at least two more data points for $T > T_{CDW}$ to reveal the possible connection between $2 \times 2 \times 2(4)$ CDW and anisotropic resistivity.

Response: We thank the reviewer for the appreciation of our work, and the constructive suggestions. Thanks to the well protected status of electrodes and contacts on sample 2, we succeed in redoing the measurements and thus add extra data on sample 2 at temperatures above T_{CDW} , see Fig. 5d and Supplementary Fig. 12. However, the status of the electric connection of sample 1 has degraded, thus we re-cleave sample 1 in order to get fresh surfaces and carry out c -axis resistivity measurements in re-cleaved sample 1. The newly obtained data of sample 1 are added to Supplementary Figs. 6, 9 as well as the newly added Supplementary Fig. 13. The data obtained in re-cleaved sample 1 are consistent with those measured in the original sample 1, and the normal-state nematicity also disappears near T_{CDW} . In addition, we add the semilog plots of Fourier transformation results obtained at high temperatures to Supplementary Fig. 8d,f. One can see clearly that the amplitude of the 180° (or C_2) peak is reduced to the background signal when the temperature is increased to that near T_{CDW} , and the amplitude is always in the noise level at high temperatures above T_{CDW} . We also add the related descriptions to the Discussion part in the main text. We hope the reviewer will be satisfied with the revised version.

Responses to the Report of Reviewer #3

The authors have properly addressed the points I previously raised. This observed interesting twofold symmetry of c -axis resistivity will stimulate further studies on the correlated phenomena in kagome metal CsV_3Sb_5 . I can agree with this paper being published in Nature Communications if other referees and the editors agree.

Response: We thank the reviewer for the appreciation and the support of publication of our manuscript.

REVIEWERS' COMMENTS

Reviewer #1 (Remarks to the Author):

The authors properly addressed my concerns and now I'm happy to recommend it to publish Nature Communications.